

# The impact on life cycle carbon footprint of converting from disposable to reusable sharps containers in a large US hospital geographically distant from manufacturing and processing facilities

Brett McPherson[1], Mihray Sharip[1] and Terry Grimmond[2]

[1] Environmental Health, Loma Linda University Health, San Bernardino, CA, United States of America
[2] Grimmond and Associates, Microbiology Consultancy, Hamilton, New Zealand

Corresponding author
Terry Grimmond,
terry@terrygrimmond.com

## ABSTRACT

**Background**. Sustainable purchasing can reduce greenhouse gas (GHG) emissions at healthcare facilities (HCF). A previous study found that converting from disposable to reusable sharps containers (DSC, RSC) reduced sharps waste stream GHG by 84% but found transport distances impacted significantly on GHG outcomes and recommended further studies where transport distances are large. This case-study examines the impact on GHG of nation-wide transport distances when a large US health system converted from DSC to RSC.

**Methods**. The study's scope was to examine life cycle GHG emissions during 12 months of facility-wide use of DSC and RSC at Loma Linda University Health (LLUH). The facility is an 1100-bed US, 5-hospital system where: the source of polymer was distant from the RSC manufacturing plant; both manufacturing plants were over 3,000 km from the HCF; and the RSC processing plant was considerably further from the HCF than was the DSC disposal plant. Using a "cradle to grave" life cycle GHG tool we calculated the annual GHG emissions of $CO_2$, $CH_4$ and $N_2O$ expressed in metric tonnes of carbon dioxide equivalents ($MTCO_2eq$) for each container system. Primary energy input data was used wherever possible and region-specific energy-impact conversions were used to calculate GHG of each unit process over a 12-month period. The scope included Manufacture, Transport, Washing, and Treatment & disposal. GHG emissions from all unit process within these four life cycle stages were summed to estimate each container-system's carbon footprint. Emission totals were workload-normalized and analysed using $CHI^2$ test with $P \leq 0.05$ and rate ratios at 95% CL.

**Results**. Converting to RSC, LLUH reduced its annual GHG by 162.4 MTCO2eq ($-65.3\%$; $p < 0.001$; RR 2.27–3.71), and annually eliminated 50.2 tonnes of plastic DSC and 8.1 tonnes of cardboard from the sharps waste stream. Of the plastic eliminated, 31.8 tonnes were diverted from landfill and 18.4 from incineration.

**Discussion**. Unlike GHG reduction strategies dependent on changes in staff behavior (waste segregation, recycling, turning off lights, car-pooling, etc), purchasing strategies can enable immediate, sustainable and institution-wide GHG reductions to be achieved. This study confirmed that large transport distances between polymer manufacturer, container manufacturer, user and processing facilities, can significantly impact the carbon footprint of sharps containment systems. However, even with large transport

distances, we found that a large university health system significantly reduced the carbon footprint of their sharps waste stream by converting from DSC to RSC.

## INTRODUCTION

Healthcare activities account for 5.4% of greenhouse gas (GHG) emissions in the U.K. (*NHS, 2016*; *DBEIS, 2017*) and 9.8% in US (*Eckelman & Sherman, 2016*) and, in hospitals, more than half of GHG emissions are derived from supply chain goods and services (*NHS, 2017*). Many hospitals are adopting green purchasing strategies to reduce their GHG (*Chung & Meltzer, 2009*; *NHS, 2017*)—a position supported by the Alliance of Nurses for Health Environments (*ANHE, 2017*). Replacing disposable products with reusables is such an example (*WHO, 2009*; *Unger et al., 2016*; *Karlsson & Ohman, 2005*) and, as clinical waste containers are in the top 20 contributors to the supply chain carbon footprint (*NHS, 2017*), replacing disposable sharps containers (DSC) with reusable sharps containers (RSC) is recommended (*PGH, 2013*). One life cycle carbon footprint study found that converting from DSC to RSC achieved a significant reduction in GHG however the authors' sensitivity analysis found transport distances could significantly affect results and, given the hospital was close to where both containers were manufactured, recommended that scenarios with large transport distances be investigated (*Grimmond & Reiner, 2012*). Our case-study compares the annual impact on life cycle carbon footprint of converting from DSC to RSC at a large US teaching hospital system sited at nationwide distances from manufacturing plants.

## MATERIALS AND METHODS
### Study overview
The scope of the study was to examine the life cycle carbon footprint of DSC and RSC over a 12-month period of facility-wide usage at a hospital geographically distant from manufacturing and processing plants, and include all unit processes in Manufacture, Transport, Washing, and Treatment & disposal stages.

Using established principles for assessment of the life cycle GHG emissions of goods and services (*British Standards Institute, 2011*) we utilised a cradle-to-grave life cycle inventory (LCI) and a product-system GHG assessment tool developed specifically for sharps containers and containing some 750 data cells (*Grimmond & Reiner, 2012*). In a before-after intervention study using a calculation model, we compared the annual GHG emissions for facility-wide usage of DSC and RSC at Loma Linda University Health (LLUH). The facility is an 1100 bed university healthcare system with 5 general acute care hospitals and an expansive outpatient clinic system in Loma Linda, California. The GHG included were $CO_2$, $CH_4$ and $N_2O$ as these represent more than 99.5% of $CO_2$eq generated

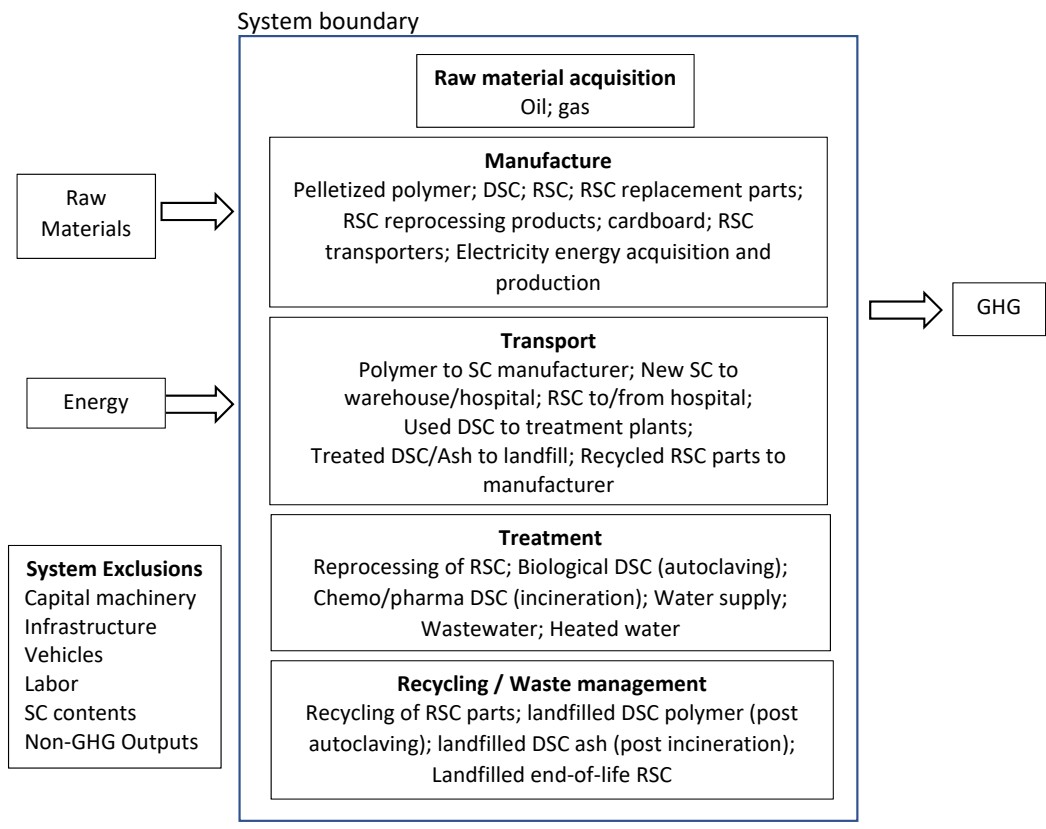

**Figure 1  System boundary showing inputs, outputs, inclusions and exclusions.**

during the major life cycle stages of sharps containers (*American Chemistry Council, 2010*; *EPA, 2016a*). Greenhouse gases, other than CO2, were converted to their CO2eq on the basis of their per unit radiative forcing using 100-year global warming potentials defined by the Intergovernmental Panel on Climate Change (*British Standards Institute, 2011*). The annual GHG emissions for each container's life cycle were expressed in metric tonnes of carbon dioxide equivalents (MTCO$_2$eq). All GHG data sources used in the study provided GHG outcomes as MTCO2eq. Review Board approval by LLUH was waived as no patients, patient data or patient specimens were involved.

The LCI itemised all energy-using processes required by each containment system's life-cycle as implemented at LLUH. Scope 1, 2 and 3 processes were included in both study years. Unit process GHG were collated into the following life-cycle stages: manufacture (of polymer and containers); transport; washing (RSC); and treatment & disposal. We assessed GHG emissions from all energy used in these processes (vehicle fuel, gas, electricity, water supply and treatment) and in the manufacture and life cycle of ancillary products (pallets, transport cabinets, cardboard boxes, wash products). The boundary of the system studied, together with inputs, outputs and exclusions, are shown in Fig. 1.
## Data sources

The following data sources were used in calculating GHG: DSC and RSC resin manufacture (*American Chemistry Council, 2010*); primary energy input data for DSC and RSC container manufacture (*Clarion, 2014*) and RSC washing (*Daniels, 2017*); industry-specific data for DSC autoclaving (*Daniels, 2012*); RSC and DSC transport (*DEFRA, 2015*); eGRID values for California, Michigan and Illinois power generation (*EPA, 2016b*); National data for energy inputs for US water supply and treatment (*Chini & Stillwell, 2018*); Industry data for manufacture of wash products (*Nielsen, Li & Zhang, 2013*; *Shahmohammadi et al., 2017*); Industry data for manufacture of cardboard (*NCASI, 2017*), representative data for manufacture of transporters (*USDOE, 2010*); Industry-specific data for pallet life cycle GHG (*DEFRA, 2010*): and US national values for incineration of DSC (*EPA, 2018*). The same database and values were applied to the relevant unit processes in DSC and RSC systems. Emissions for RSC manufacturing were calculated using a worst-case scenario based on the actual age of the manufacturer's oldest, most frequently used RSC still in service nationally. Although it is theoretically possible for RSC to be recertified for a further period when they reach their certified reuse expiration, for this study their "end-of-life" was conservatively taken to be the number of years under the above worst-case scenario. The GHG associated with manufacture of ancillary reusables (transport-cabinets and pallets) were calculated on a per trip basis using their expected life span. Data on container size, model number, number used, and total Adjusted Patient Days (APD) (workload indicator) were obtained from LLUH. Total polymer required for manufacture of DSC and RSC was determined by weighing an example of each model of container and multiplying by the number of containers. The conversion-transition period (2 years) was excluded to avoid system overlap. Emission totals for each system's annual use were workload-normalized by dividing its life cycle MTCO2eq by the APD for that year. The two ratios were then analyzed using WinPepi v11.65 (*WinPepi, 2016*). A Yates-corrected $\chi 2$ test was used for the analysis of proportions. Statistical significance was set at $P \leq .05$ and rate ratios calculated using 95% confidence intervals.

## System function, boundary, allocation and classification

The system function provided by the alternative products (DSC, RSC) was the supply of sharps containers for the disposal of sharps waste (biological, chemotherapeutic, pharmaceutical) within LLUH. The functional unit was the supply of each system for a one-year period. Sharps waste is a sub-category of medical waste and comprises items capable of penetrating human skin (e.g., needles, scalpels) which may have the potential to transmit infectious disease or pose a physical or chemical hazard. Because of these hazards, at disposal, all sharps must be safely contained in either DSC or RSC and transported to a treatment facility. With DSC the container is used once and the intact container and contents are subjected to treatment (commonly autoclaving, or incineration) prior to landfill. With RSC, the container is automatedly decanted of its contents (which are treated and disposed), and the reusable container is robotically cleaned and decontaminated, and reused a defined number of times. The boundary of the system studied (Fig. 1) included the energy required for the following unit processes: raw material extraction;

polymer manufacture and transport; container manufacture and transport; transport of full containers to treatment facility; RSC processing-energy (including water supply, water treatment, and wash products); treatment of DSC; transport of treated DSC to landfill; and energy required for electricity generation and supply. Transport fuel processes were calculated from well to wheel. Excluded from the system boundary were treatment of container contents (identical in both DSC and RSC), infrastructure and assets, and any inputs and outputs that comprised less than 1% of mass or energy (*British Standards Institute, 2011*), or were not relevant to carbon footprint.

The production of polymer from oil or gas is a multi-function process and allocation of emissions and resource use was performed on a mass basis, as was transport, autoclaving and pallet manufacture. The injection-molding of DSC/RSC and the processing (cleaning and decontamination) of RSC are single-function processes and no allocation to co-products was necessary. Incineration of chemotherapeutic and pharmaceutical DSC was carried out in waste to energy incinerators that co-produce electricity and the avoided utility emissions were subtracted to give net GHG emissions per ton of specific polymer incinerated (*EPA, 2018*). In cardboard production (1.7% of total DSC life-cycle GHG) allocation was averaged using cut-off and number-of-uses methods where appropriate (*NCASI, 2017*). Regional emissions reported in eGrid represent electricity generation only—any emissions used for purposes other than making electricity were excluded from the adjusted emissions (*EPA, 2016b*).

Global warming was the impact assessment category to which all inventory data was classified as it is well-known and commonly used and understood by healthcare facilities. A table listing the raw data for all unit processes including flow, units, conversion factors, total GHG, data sources and data-representativeness, accompanies this publication.

## RESULTS

DSC were manufactured in Crystal Lakes IL from US-sourced polypropylene polymer, nested in cardboard containers, transported 3,200 km to the hospital on wooden pallets, and autoclaved and landfilled without shredding at Vernon CA, 130 km from the hospital. The RSC were manufactured in Greenville MI from polymer sourced in Korea, transported 3,500 km in reusable, proprietary transporter cabinets to LLUH, and decanted and processed at Fresno CA, 440 km from the hospital.

A summary of results is presented in the Table 1.

To service LLUH in the baseline year, 48,460 DSC were manufactured from 50.6 tonnes of polymer and required 8.2 tonnes of corrugated cardboard packaging for transport (see Table 1). The DSC used did not contain recycled polymer. In California, biological sharps are treated by non-incineration technologies (e.g., autoclave) then landfilled; chemotherapeutic and pharmaceutical sharps must be incinerated (and ash landfilled)—this requires transport interstate as there are no licensed incinerators for such wastes in California. With DSC, this resulted in 31.8 tonnes of plastic DSC being landfilled and 18.8 tonnes of DSC being transported interstate for incineration  (Table 1).

In the RSC year, 2,779 RSC were manufactured from 9.6 tonnes of acrylonitrile butadiene styrene (ABS) polymer, and 0.4 tonnes of cardboard were used for packaging of 412 chemo

Table 1 **Annual sharps waste stream and GHG: comparison of disposable vs reusable sharps containers at LLUH.**

|  | DSC | RSC |
|---|---|---|
| Containers manufactured | 48,460 | 3,195[a] |
| Containers landfilled annually | 35,925[b] | 0[c] |
| Weight plastic landfilled (tonnes) | 31.8 | 0[c] |
| Weight plastic incinerated (tonnes) | 18.8 | 0.4[d] |
| Weight cardboard boxes (tonnes) | 8.2 | 0.1[e] |
| Container exchanges | 48,460 | 33,356[f] |
| MTCO2eq GWP[g] | 248.62 | 86.19 |
| Adjusted Patient Days (APD) | 296,205 | 297,056 |
| MTCO$_2$eq GWP per 10,000 APD[h] | 8.37 | 2.90[i] (−65.3%) |

**Notes.**

GHG, Greenhouse Gas; LLUH, Loma Linda University Health; MTCO$_2$eq, metric tonnes carbon dioxide equivalent; DSC, disposable sharps container; RSC, reusable sharps container; GWP, Global Warming Potential.

[a]2,779.7 RSC manufactured in year one only, plus 3.7 replacement RSC annually (allowing for reuse and recycling credits), plus 412 chemotherapy/pharmaceutical DSC annually.

[b]8,245 Chemotherapy/Pharmaceutical DSC were incinerated/yr.

[c]No RSC were landfilled as all parts were either reused or recycled.

[d]Tonnes of chemo/pharma DSC incinerated (412 chemo DSC were used during RSC year).

[e]Chemotherapy DSC packaging.

[f]RSC were larger in fill-line capacity (25.7L vs DSC 18.5L) and exchanged less often than DSC.

[g]Emissions of GHG expressed in terms of global warming potentials, defined as the radiative forcing impact of one mass-based unit (kg) of a given GHG relative to an equivalent unit of carbon dioxide over a given period of time (100 years) (*British Standards Institute, 2011*).

[h]10,000 APD used as workload denominator to normalize base year comparison and facilitate inter-hospital comparisons.

[i]65.3% reduction; $P < 0.001$; Rate Ratio = 2.90; CL(95%) = 2.27–3.71.

DSC that were continued to be used (no cardboard is used for RSC packaging). During the RSC study year, approximately 60 RSC required repair with 30 kg parts being recycled (80%) or reused (20%) (nil to landfill), and, with recycling credit, an equivalent of 3.7 RSC were manufactured as replacement containers (2,783 RSC total for year). In the RSC study-year, the manufacture, treatment and disposal of 412 chemotherapy DSC were included. The RSC in this study, certified for 500 uses, were reused an average of 12.0 times/year at LLUH, giving a theoretical "end-of-life" lifespan of 41.7 years. However a "worst-case" lifespan scenario was adopted based on manufacturer's data on the number of reuses of the most frequently used RSC still in service in the US (each individual RSC is barcoded and its uses monitored). The manufacturer stated their oldest and most frequently used RSC still in service in the US was 19 years old and had been used 360 times, thus giving a "worst-case" lifespan of 26.4 years for this container. Manufacturing GHG for RSC (calculated by dividing total manufacturing GHG by life expectancy) was 1,135 kg CO2eq for a lifespan of 41.7 years (1.3% of total RSC life-cycle GHG) and 1,795 kg CO2eq for a worst-case lifespan of 26.4 years (2.1% of total RSC life-cycle GHG). The shorter, worst-case lifespan was used in this study. Total GHG emissions and GHG differences between DSC and RSC life cycle stages are shown in Fig. 2.

Adjusting for the 0.3% APD workload increase in the year of RSC use, sharps management GHG using DSC was 248.6 MTCO2eq, and with RSC use, decreased to

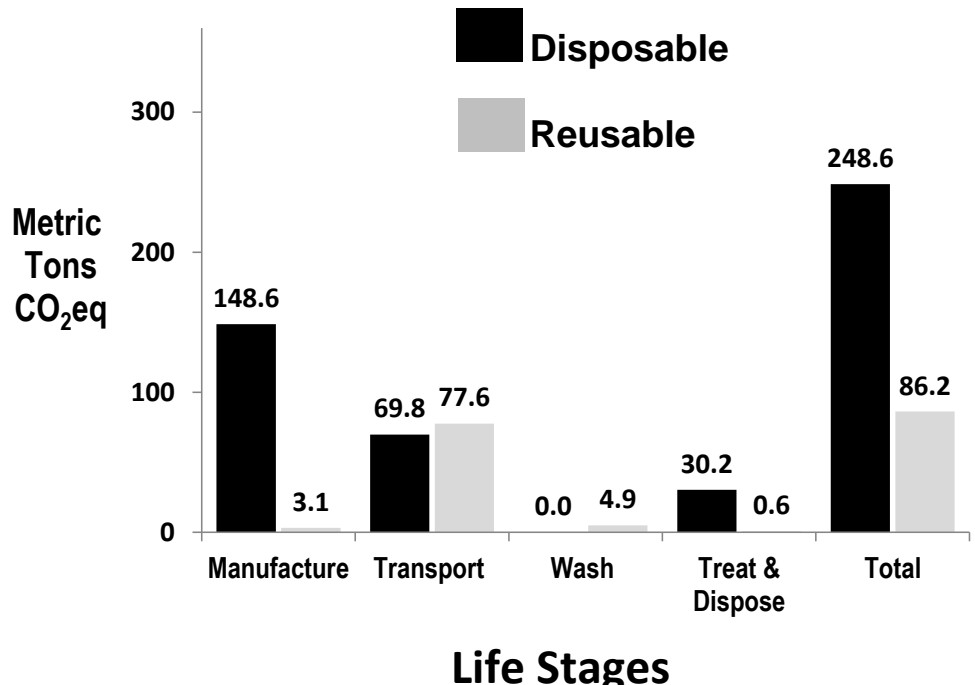

**Figure 2** Annual greenhouse gas emissions by life stage of disposables and reusable sharps containers at Loma Linda University Hospital, with DSC normalised to Adjusted Patient Days.

86.20 MTCO2eq, a 162.4 MTCO2eq reduction in carbon footprint (65.3%, $p < 0.001$, RR 2.27–3.71) (see Table 1 and Fig. 2).

In addition to the GHG reduction with RSC, LLUH annually eliminated 50.2 tonnes of plastic DSC and 8.1 tonnes of cardboard from the sharps waste stream. Of the plastic eliminated, 31.8 tonnes were diverted from landfill and 18.4 from incineration.

## DISCUSSION

### Background and impact of distances

Commercial RSC, first used in US and Australia in 1986, now represent approximately 50% and 75% respectively of the sharps containers used in these countries, and since 1999 have been increasingly used in Canada, UK, Ireland, New Zealand, South Africa and South America. Generally, RSC are reused many times per year and, with rugged construction and effective inspection and repair, may last several decades. Prior to marketing in the U.S., RSC and DSC are required by the US Food and Drug Administration (FDA) to pass identical performance tests and design requirements as stipulated in sharps container standards (*FDA, 1993*). However, prior to this testing, FDA require RSC to undergo "lifespan simulation" and suggest

1. The containers be filled & processed for the number of lifespan uses stated by the manufacturer (e.g., 500 times); then,

2. the same containers be subjected to a transport vibration test, e.g., US Department of Transport Packaging Vibration Standard (*USDOT, 2001*), and then,

3. the same containers must pass the tests and performance criteria of a Sharps Container Standard.

Likewise, the Canadian sharps container standard does not distinguish between DSC and RSC in its performance test requirements and requires lifespan simulation of RSC prior to testing (*CSA, 2014*).

One reason healthcare facilities adopt RSC is for environmental sustainability (*PGH, 2013*) but quantitative studies confirming this fact are rare (*Unger et al., 2016*; *Karlsson & Ohman, 2005*).

A government study in the UK confirmed medical waste containers are among the top 20 items that account for more than 70% of the supply chain footprint and, to reduce the footprint, recommended: manufacturers report footprints of their products; reductions in quantity purchased; and sourcing of low carbon alternatives; (*NHS, 2017*). Our study found that converting from DSC to RSC significantly reduced the carbon footprint, and eliminated 50.2 tonnes of plastic and 8.1 tonnes of cardboard from the sharps waste stream.

Although the same RSC may be reused several hundred times, energy is required for their robotic washing between uses and, being heavier than DSC, their greater weight means more energy is required per unit for transport and manufacture. *Ali, Weng & Chaudhry (2017)* noted that GHG increase considerably when medical waste is transported longer distances. A previous life cycle study found that when container-manufacturing plants and RSC processing plant are close to the healthcare facility (HCF), the conversion to RSC resulted in an 83.5% reduction in GHG, and transport contributed 25.8% to the RSC life-cycle GHG (*Grimmond & Reiner, 2012*). In our study, the HCF was 3,500 km from the RSC manufacturing plant, and, more importantly (because of daily delivery), the RSC processing plant was 440 km from the HCF. This resulted in transport GHG accounting for 90.6% of the RSC life-cycle GHG (see Fig. 2). However, notwithstanding that these longer distances lessened the GHG differential between DSC and RSC, the conversion to RSC significantly reduced total sharps waste management GHG by 65.3%. The reduced number of container exchanges with RSC (with associated labor reduction) was also noteworthy (Table 1). The reduction in sharps management GHG with RSC use, while only a small component of the total supply chain emissions at LLUH, has been a positive step in the institution's sustainability strategies. Unlike GHG reduction strategies dependent on changes in staff behaviour (waste segregation, turning off lights, car-pooling, etc.), our study confirms that purchasing strategies can enable immediate, sustainable and institution-wide GHG reductions to be achieved.

## Impact on GHG over 10 years

The impact of repeated DSC manufacture and one-off RSC manufacture is best illustrated over multiple years. In the LLUH scenario over a 10-year period, 484,600 DSC would need be manufactured compared to 2783 RSC (and 4,120 chemo DSC), and would divert 502 tonnes of plastic from landfill or incineration.

## Sensitivity analysis

Manufacturing (of polymer and containers) gave the largest differential between the two systems (see Fig. 2) and is predominantly a function of the energy required for the higher total polymer weight needed to be annually manufactured and molded for DSC. Although more DSC required transportation from the distant manufacturing plant, the daily transport of RSC from the distant processing plant resulted in a similar transportation GHG for both systems over the year (see Fig. 2). The sensitivity analysis revealed that variations in RSC lifespan contributed little to the GHG result—reducing RSC lifespan from a theoretical 41.7 years to 26.4 years (used in this study) or 15 years, reduced the DSC:RSC GHG difference by only 0.4%, and 1.3% respectively.

Electricity "cleanliness" across US grids (e.g., wind, coal, hydro) is a key variable in comparative GHG analyses (*Unger et al., 2016*) and the sensitivity analysis in our study showed that differing US electricity sources can alter processing and manufacturing GHG by 82% which, when extrapolated to the total life-cycle, can alter DSC GHG by 23% and RSC GHG by 10%. Optimization of reprocessing of medical products is recommended to lower GHG (*Unger et al., 2016*) however, in this scenario, RSC reprocessing accounted for only 5.6% of total RSC life-cycle GHG. Our analysis confirmed findings of other studies (*Grimmond & Reiner, 2012*; *Unger et al., 2016*), that material reclamation could reduce DSC life-cycle GHG if reclaimed plastic is used to offset virgin polymer use.

## Other impacts of RSC

The focus of this study was carbon footprint however cost reduction (*Grimmond & Reiner, 2012*) and sharps injury reduction (*Grimmond et al., 2010*) have also been associated with RSC use and these factors, together with sustainability and mandatory frontline staff evaluation, were considered prior to adoption of the RSC system by LLUH. In terms of environmental impacts, we considered only one, global warming, however other impact categories such as ozone depletion, ecotoxicity, acidification, particulate matter, eutrophication, and human toxicity, may enable additional conclusions to be drawn (*Eckelman & Sherman, 2016*).

## Study limitations and strengths

One limitation of the study was the assumption made in the location of manufacture of polymer for the DSC. To limit the impact of this assumption, the location was conservatively assumed to be a United States polymer-supplier close to the point of manufacture of the DSC. A second limitation was the use of the UK DEFRA database for transport energy inputs. This was necessary as no relevant United States database using tonne.km was available; however, all databases were applied equally to DSC and RSC systems. Study strengths were in the availability of 12 months of detailed usage data for both systems; the large transport distances compared to previous studies; the use of a conservative RSC lifespan; and the primary and region-specific availability of energy input data for unit processes in both systems.

## CONCLUSIONS

- Large RSC transport distances lessen the differential between DSC and RSC GHG, however, RSC still achieved significant GHG reductions over DSC.
- Transport & electricity cleanliness are key factors in GHG of sharps waste management.
- RSC lifespan has minimal effect on carbon footprint comparisons of container-types.
- Purchasing decisions can significantly contribute to HCF GHG-reduction strategies.
- Institution-wide adoption of RSC can reduce GHG with minimal staff behavior-change.

### Funding

Daniels Health granted $2500 towards the cost of the study, which covered approximately 10% of expenses. No other grant or funding was received from any funding agency in the public, commercial, or not-for-profit sectors. The funders had no role in study design, data collection and analysis, decision to publish, or preparation of the manuscript.

### Grant Disclosures

The following grant information was disclosed by the authors:
Daniels Health.

### Competing Interests

Brett McPherson and Mihray Sharip declare no conflict of interest. Terry Grimmond is an international consultant in sharps injury prevention and waste management to healthcare and associated industries. Daniels Health, the manufacturer of the reusable device studied in this paper, is one of his clients. The manufacturer did not review, sight or have input into the design, content, methodology, results, write-up of the study or choice of journal for publication.

### Author Contributions

- Brett McPherson conceived and designed the experiments, performed the experiments, contributed reagents/materials/analysis tools, authored or reviewed drafts of the paper, approved the final draft, sought IRB approval.
- Mihray Sharip conceived and designed the experiments, performed the experiments, contributed reagents/materials/analysis tools, authored or reviewed drafts of the paper, approved the final draft.
- Terry Grimmond conceived and designed the experiments, performed the experiments, analyzed the data, prepared figures and/or tables, authored or reviewed drafts of the paper, approved the final draft.

### Data Availability

Raw data are provided in the Supplemental Materials.

## Supplemental Information

Supplemental information for this article can be found online at http://dx.doi.org/10.7717/peerj.6204#supplemental-information.

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
