# Peer review of "The impact on life cycle carbon footprint of converting from disposable to reusable sharps containers in a large US hospital geographically distant from manufacturing and processing facilities"

_PeerJ, doi:10.7717/peerj.6204_

## Round 0.1 · original submission · Major Revisions

Dear Dr Grimmond

You have Appealed the previous Rejection decision, and I have been asked to handle the Appeal. I have reviewed your appeal letter, dated 7th March 2018, regarding the rejection decision you received and I believe that the grounds you presented firmly address the reviewer and previous Editor’s issues. Therefore, I agree that your paper can be subjected to another revision and review stage.

I am thus giving this submission a 'Major Revision' decision and I invite you to submit your revision and rebuttal materials, at which point I will take it forward for further peer-review.

Reviewer 1 ·

Basic reporting

No comment

Experimental design

Using an anticipated life expectancy of 41.7 years for reusable sharps containers seems unrealistically long in a high use clinical unit e.g. operating theatre, oncology / chemotherapy ward. I would be interested to see calculations using a more realistic life expectancy e.g. 5 years, 10 years, 15 years.

Validity of the findings

No commnet

Additional comments

no comment

---

## Round 0.2 · Major Revisions

Further to your Appeal, we re-reviewed this manuscript.

Please address all the raised comments by the reviewers carefully with particular reference to the reviewers 2 and 4 and underline your point-by-point response to these comments. We look forward to receiving your revision.

·

Basic reporting

This paper addresses the environmental implications of changing from disposable to reusable sharps containers. This is a relevant topic, because sustainability in a hospital context is an underdeveloped theme. The study uses life cycle assessment (LCA) as a method, which is appropriate because there is a trade-off between the impacts of production and treatment from disposable containers and the impacts of decontaminating from reusable containers. Moreover, PeerJ is an important journal in the medical world, so messages broadcasted through PeerJ have the potential to be received, and may change hospital practice. As such, I appreciate the manuscript.

From another side, there is much to comment on. I have 25 years experience in the field of LCA, and the paper violates many basic rules of good LCA practice. The standard for clear reporting as outlined in the ISO standards 14040-14044 is not followed. This implies that many important choices and issues are addressed in a limited or obscure way. For instance, "allocations for emissions were as follows ... electricity generation (kWh); water supply and treatment (litre)": I have no idea how these allocations were done in detail, and I would have difficulty in reproducing the study. The purpose of the ISO standards is to make studies transparent and reproducible. Another example: I have no idea which greenhouse gases were included (CO2, CH4, N2O, etc). The online supplementary info states a "GWP emission factor" for different plastics and transport activities, a highly unusual way of reporting, given the widely-endorsed preference of separating inventory analysis and impact assessment. A third example: it is a usual practice to include uncertainty analysis, offering ranges or distribution of outcomes rather than point values. Most LCA-software nowadays features Monte Carlo analysis for this.

The paper does offer some statistics, on line 174, but this p-value doesn't seem to be based on repeated sampling. Or is it? I do understand the 65,3% reduction, but I have no clue how this translates into p-value or an RR. Perhaps this is a good novelty, but it is uncommon in LCA, so please report how it was done and what it means.

Line 244 discusses "other impacts" besides global warming (by the way, the impact is global warming, not global warming potential or GWP; GWP is calculated by IPCC, not by you). But it doesn't mention other environmental impacts. LCA studies should not be restricted to global warming, but usually include acidification, toxic emissions, and many more impacts. Please check http://eplca.jrc.ec.europa.eu/uploads/ILCD-Handbook-LCIA-Framework-Requirements-ONLINE-March-2010-ISBN-fin-v1.0-EN.pdf or another contemporary document on LCA. Inclusion of a broader range of impacts mught affect the results quite a bit, either strenghtening the conlusion or weakening it.

Experimental design

Basically, no experiments were done, this work is based on a calculation model (LCA).

Validity of the findings

The validity is difficult to check; see my points made above.

·

Basic reporting

no comment

Experimental design

no comment

Validity of the findings

no comment

Additional comments

Overall the article is well structured, the experiments and analysis are appropriate, the discussion and conclusion are solid and convincing. I have one question on a tiny detail in the article

line 163 to line 167 discusses the 'worst-case' lifespan of RSCs, from the evidence it seems in the 'worst-case' scenario, the RSCs are used 360 times in 19 years, that's 360/19 = 17 times/year, which is above the average of 12 times/years (line 162). My question is

How does the lifespan is correlated to the reuse times per year?
and
How does 17 times/year accountable for the worst case?

This question doesn't affect my decision, it's up the author to answer this question for me.

Reviewer 4 ·

Basic reporting

Some of the information in the introduction is outdated. US GHG emissions e.g. Chung and Meltzer US GHG estimate of 7.6% in 2009 has been updated by Eckelman and Sherman to 9.8% in 2016 publication, and should be updated to the most recent. Further, while the NHS publication notes more than half of GHG emissions stem from procurement, this is not reflected in either Chung or Eckelman since the data is aggregated differently. Thererefore, please remove the Chung citation from line 53.

Abstract/discussion: “..sharps wastes….can account for up to half of total medical waste” is not supported by the citations noted. Either support this claim or remove.

Lines 181-183 require an original citation.

Experimental design

In terms of waste management, for sharps without pharmaceuticals, treatment is not necessarily simply autoclaving and landfilling. Rather such sharps waste must be rendered both non-infectious and non-recognizable (i.e., no longer sharp) in several states (and I believe this includes California where this study takes place). The latter treatment is at higher temperature than standard autoclaving. The authors should clarify and cite their state regulations, and correct their calculations if necessary.

Validity of the findings

Original data and reproducibility acceptable.

Additional comments

Overall, this study is well done, but is VERY similar to the study, Grimmond T and Reiner S, 2012 study. The only difference is that this study is performed at a different hospital in a different state, thus the transportation and waste volumes are different.

---

## Round 0.3 · Minor Revisions

We are pleased to send you the review comments, included at the bottom of this letter. Please consider these suggestions. We advise you to address all the raised comments carefully and underline your point-by-point response to these comments.

We look forward to receiving your revision.

·

Basic reporting

See below

Experimental design

NA

Validity of the findings

NA

Additional comments

The authors have addressed a lot of my earlier comments mainly by more explicitly positioning the article not as an LCA but as carbon footprint. This would allow them to focus the study on greenhouse gases (GHG) and at the same time to disregard the ISO-standards and comply only with the BSI-standard PAS 2050. True as this may sound, there are two important remarks to be made.
1) With an exclusive emphasis on GHG and climate change, other important types of environmental impact are out. The section "Other impacts of RSC" very briefly addresses other impacts, but these are only cost and sharps injury. Through this, a quick reader can think that RSC are from an environmental point of view superior to DSC. The authors should add a longer text in the "Other impacts" section to inform the reader that there is more in the environment than GHG only, such as ozone layer, toxic emissions, acidifying emissions, water withdrawal, fossil resources, etc etc, but that this study only looked at GHG, so that a full LCA might lead to different conclusions, or might strengthen the preference for RSC, we simply don't know.
2) ISO-LCA allows for a smaller subset of impacts to be considered, when motivated. In that sense, a carbon footprint might still be ISO-LCA-compliant. BSI-PAS 2050 is very much based on ISO-LCA, see e.g. PAS 2050 clause 4.1 "Assessment of the GHG emissions of products shall be carried out using LCA techniques". This includes explicitly the issue of "Transparency: where the results of life cycle GHG emissions assessment carried out in accordance with this PAS are to be disclosed to a third party, GHG emissions-related information is made available that is sufficient to support disclosure and allow such a third party to make associated decisions with confidence". While I agree that it is not easy to create a fully transparent report within the constraints of an article, the present text still leaves out many details or gives a confused presentation. One example: I can hardly believe that "no significant allocation issues were considered", because there is electricity, refineries and carton in your system, and for these items allocation can be a decisive factor. Another example: the online supplementary information contains a "CO2eq Characterization Factor (CF)" for "polypropylene manufactured for DSC Study year", which is plainly wrong. Characterization factors exist for CH4, N2O, but not for polypropylene; check IPCC. Third example: the DSC is incinerated after disposal, but probably with energy recovery, and it is not clear if any recovered enrgy bonus was given for this.
In conclusion, the move from LCA to carbon footprint is a bit too cheap now.

---

## Round 0.4 · accepted · Accept

Thank you for your contribution to PeerJ and we look forward to receiving further submissions from you.